# TRANSFORMATION AUTOREGRESSIVE NETWORKS

## ABSTRACT

The fundamental task of general density estimation has been of keen interest to machine learning. Recent advances in density estimation have either: *a*) proposed a flexible model to estimate the conditional factors of the chain rule, $p(x_i \mid x_{i-1}, \ldots)$; or *b*) used flexible, non-linear transformations of variables of a simple base distribution. Instead, this work jointly leverages transformations of variables and autoregressive conditional models, and proposes novel methods for both. We provide a deeper understanding of our methods, showing a considerable improvement through a comprehensive study over both real world and synthetic data. Moreover, we illustrate the use of our models in outlier detection and image modeling tasks.

## 1 INTRODUCTION

Density estimation is at the core of a multitude of machine learning applications. However, this fundamental task, which encapsulates the understanding of data, is difficult in the general setting due to issues like the curse of dimensionality. Furthermore, general data, unlike spatial/temporal data, does not contain a priori known correlations among covariates that may be exploited and engineered with. For example, image data has known correlations among neighboring pixels that may be hard-coded into a model, whereas one must find such correlations in a data-driven fashion with general data.

In order to model high dimensional data, a large number of methods have considered auto-regressive models, which model the conditional factors of the chain rule (Uria et al., 2013; 2016; Germain et al., 2015; Gregor et al., 2014). I.e these models estimate the conditionals: $p(x_i|x_{i-1}, \ldots, x_1)$, for $i \in \{1, \ldots, d\}$. While some methods directly model the conditionals $p(x_i|x_{i-1}, \ldots)$ using sophisticated semiparametric density estimates, other methods apply sophisticated transformations of variables $x \mapsto z$ and take the conditionals over $z$ to be a restricted, often independent base distribution $p(z_i|z_{i-1}, \ldots) \approx f(z_i)$ (Dinh et al., 2014; 2016; Goodfellow, 2016). In this paper we take a step back from these previous approaches that have considered either: *a*) a flexible autoregressive scheme with simple or no transformations of variables (Figure 1a); or *b*) a simple autoregressive scheme with flexible transformations of variables (Figure 1b) . We leverage both of these approaches (Figure 1c), develop novel methods for each, and show a considerable improvement with their combination.

**Contributions:** The following are our contributions. First, we propose two flexible autoreggressive models for modeling conditional distributions: the linear autoregressive model (LAM), and the recurrent autoregressive model (RAM). LAM employs a simple linear form to condition on previously seen covariates in a flexible fashion. RAM uses a recurrent neural network (RNN) to evolve conditioning features as the set of conditioning covariates expands. Furthermore, this paper proposes several novel transformations of variables: 1) we propose an efficient method for learning a linear transfromations on covariates; 2) we develop an invertible RNN-based transformation that directly acts on covariates; 3) we also propose an additive RNN-base transformation. To better capture correlations in general data, we propose transformation autogressive networks (TANs) that combine our novel autoreggresive models and transformations of variables. We performed a comprehensive evaluation of autoregressive models and transformations that shows the fundamental result that modern density estimation methods should employ *both* a flexible conditional model and a flexible transformation. Extensive experiments on both real and synthetic datasets show the power of TANs for capturing complex dependencies between the covariates. Moreover, we show that the learned model can be used for outlier detection, and image modeling.

The remainder of the paper is structured as follows. First, in Section 2.1 we present two novel methods for modeling condition distributions across covariates. Next, in Section 2.2, we describe several transformations to use in conjunction with our proposed conditional models. After, we discuss related work and contrast our approach to previous methods. We then illustrate the efficacy of our methods with both synthetic and real-world data.

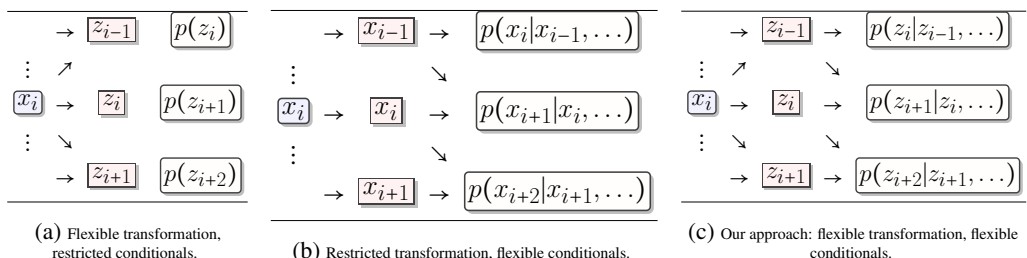

(a) Flexible transformation, restricted conditionals.

(b) Restricted transformation, flexible conditionals.

(c) Our approach: flexible transformation, flexible conditionals.

Figure 1: Illustration from left to right: (a) a flexible transformations of variables with an independent autoregressive scheme; (b) no transformations of variables with a flexible autoregressive scheme; and (c) a transformation autogressive network (TAN) that has a flexible transformation and a flexible autoregressive scheme.

## 2 TRANSFORMATION AUTOREGRESSIVE NETWORKS

First, we propose two autoregressive models to estimate the conditional distribution of input covariates $x \in \mathbb{R}^d$. After, we shall show how we may use such models over a transformation $z = q(x)$, while renormalizing to obtain density values for $x$.

### 2.1 AUTOREGRESSIVE MODELS

Autoregressive models decompose density estimation of a multivariate variable $x \in \mathbb{R}^d$ into multiple conditional tasks on a growing set of inputs through the chain rule:

$$p(x_1, \ldots x_d) = \prod_{i=1}^{d} p(x_i \mid x_{i-1}, \ldots, x_1). \tag{1}$$

That is, autoregressive models will look to estimate the $d$ conditional distributions $p(x_i|x_{i-1}, \ldots)$. A particular class of autoregressive models can be defined by approximating conditional distributions through a mixture model, $\mathrm{MM}(\theta(x_{i-1}, \ldots, x_1))$, with parameters depending on $x_{i-1}, \ldots, x_1$:

$$p(x_i|x_{i-1}, \ldots, x_1) = p(x_i \mid \mathrm{MM}(\theta(x_{i-1}, \ldots, x_1))), \tag{2}$$
$$\theta(x_{i-1}, \ldots, x_1) = f(h_i) \tag{3}$$
$$h_i = g_i(x_{i-1}, \ldots, x_1), \tag{4}$$

where $f(\cdot)$ is a fully connected network that may use a element-wise non-linearity on inputs, and $g_i(\cdot)$ is some general mapping that computes a hidden state of features, $h_i \in \mathbb{R}^p$, which help in modeling the conditional distribution of $x_i \mid x_{i-1}, \ldots, x_1$. One can control the flexibility of the autoregressive model through $g_i$. It is important to be powerful enough to model our covariates while still generalizing. In order to achieve this we propose two methods for modeling $g_i$.

First, we propose the linear autoregressive model (LAM), using a straightforward linear map as $g_i$ (eq. 4):

$$g_i(x_{i-1}, \ldots, x_1) = W^{(i)} x_{<i} + b, \tag{5}$$

where $W^{(i)} \in \mathbb{R}^{p \times (i-1)}$, $b \in \mathbb{R}^p$, and $x_{<i} = (x_{i-1}, \ldots, x_1)^T$. Notwithstanding the simple form of (eq. 5), the resulting model is quite flexible as it may model consecutive conditional problems $p(x_i|x_{i-1}, \ldots, x_1)$ and $p(x_{i+1}|x_i, \ldots, x_1)$ very differently.

Next we propose the recurrent autoregressive model (RAM), which features a recurrent relation between $g_i$'s. Given the expanding set of covariates progressively fed into $g_i$'s, it is natural to consider a hidden state that evolves according to an RNN recurrence relationship:

$$h_i = g\left(x_{i-1}, g(x_{i-2}, \dots, x_1)\right) = g\left(x_{i-1}, h_{i-1}\right), \tag{6}$$

where $g(x, s)$ is a RNN function for updating one's state based on an input $x$ and prior state $s$ (see Figure 2). In the case of gated-RNNs, the model will be able to scan through previously seen dimensions remembering and forgetting information as needed for conditional densities without making any strong Markovian assumptions.

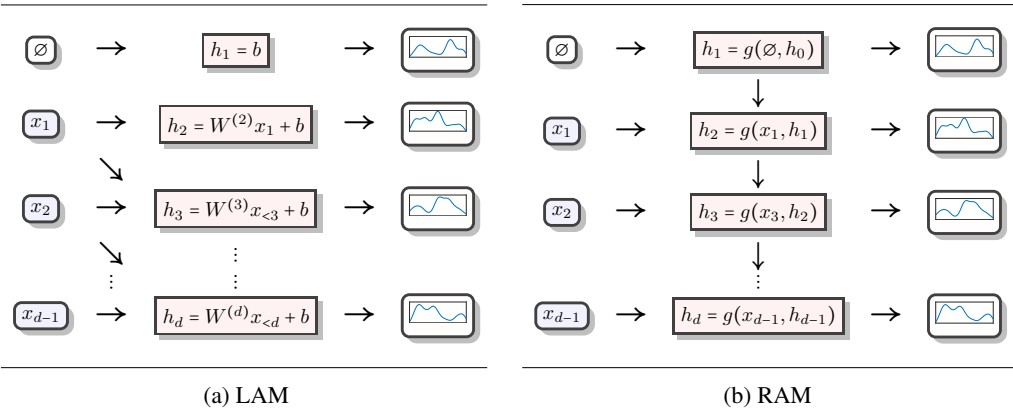

(a) LAM                     (b) RAM

Figure 2: Illustration of both LAM (left) and RAM (right) models. Hidden states $h_k$'s are updated and then used to compute the parameters of the next conditional density for $x_k$. Note that in LAM the hidden states $h_j$'s are not tied together, where in RAM the hidden state $h_j$ along with $x_j$ are used to compute the hidden state $h_{j+1}$ which determines the parameters of $p(x_{j+1} | h_{j+1})$.

Both LAM and RAM are flexible and able to adjust the hidden states, $h_i$ (eq. 4), to model the distinct conditional tasks $p(x_i | x_{i-1}, \dots)$. However, there is a trade-off of added flexibility and transferred information between the two models (see Figure 2). LAM treats the conditional tasks for $p(x_i | x_{i-1}, \dots)$ and $p(x_{i+1} | x_i, \dots)$ in a largely independent fashion. This makes for a very flexible model, however the parameter size is also large and there is no sharing of information among the conditional tasks. On the other hand, RAM provides a framework for transfer learning among the conditional tasks by allowing the hidden state $h_i$ to evolve through the distinct conditional tasks. This leads to fewer parameters and more sharing of information in respective tasks, but also yields less flexibility since conditional estimates are tied, and may only change in a smooth fashion.

## 2.2 TRANSFORMATIONS

Several methods (Dinh et al., 2014; 2016; Goodfellow, 2016) have shown that the expressive power of very simple conditional densities (eq. 1) (such as independent Gaussians) can be greatly improved with transformations of variables. Although the chain rule holds for arbitrary distributions, a limited amount of data and parameters limits the expressive power of models. Hence, we expect that combining our conditional models with transformations of variables will also further increase flexibility. When using an invertible transformation of variables $z = (q_1(x), \dots, q_d(x)) \in \mathbb{R}^d$, one renormalizes the pdf of $x$ as:

$$p(x_1, \dots x_d) = \left| \det \frac{\mathrm{d}q}{\mathrm{d}x} \right| \prod_{i=1}^{d} p\left(q_i(x) \,|\, q_{i-1}(x), \dots, q_1(x)\right), \tag{7}$$

where $\left| \det \frac{\mathrm{d}q}{\mathrm{d}x} \right|$ is a normalizing factor of the Jacobian of the transformation.

For analytical and computational considerations, we require transformations that are invertible, efficient to compute and invert, and have a structured Jacobian matrix. In order to meet these criteria we consider the following transformations.

**Linear Transformation:** First, we propose a linear transformation:

$$z = Ax + b, \tag{8}$$

where we take A to be invertible. Note that even though this linear transformation is simple, it includes permutations, and may also perform a PCA-like transformation, capturing coarse and highly varied features of the data before moving to more fine grained details. In order to not incur a high cost for updates, we wish to compute the determinant of the Jacobian efficiently. We do so by directly working over an LU decomposition $A = LU$ where $L$ is a lower triangular matrix with unit diagonals and $U$ is a upper triangular matrix with arbitrary diagonals. As a function of $L$, $U$ we have that $\det \frac{dz}{dx} = \prod_{i=1}^{d} U_{ii}$; hence we may efficiently optimize the parameters of the linear map. Furthermore, inverting our mapping is also efficient through solving two triangular matrix equations.

**Recurrent Transformation:** Recurrent neural networks are also a natural choice for variable transformations. Due to their dependence on only previously seen dimensions, RNN transformations have triangular Jacobians, leading to simple determinants. Furthermore, with an invertible output unit, their inversion is also straight-forward. We consider the following form to an RNN transformation:

$$z_i = r_\alpha \left( yx_i + w^T s_{i-1} + b \right), \quad s_i = r \left( ux_i + v^T s_{i-1} + a \right), \tag{9}$$

where $r_\alpha$ is a leaky ReLU unit $r_\alpha(t) = \mathbb{I}\{t < 0\}\alpha t + \mathbb{I}\{t \geq 0\}t$, $r$ is a standard ReLU unit, $s \in \mathbb{R}^\rho$ is the hidden state $y, u, b \, a$ are scalars, and $w, v \in \mathbb{R}^\rho$ are vectors. As compared to the linear transformation, the recurrent transformation is able to transform the input with different dynamics depending on its values. Inverting (eq. 9) is a matter of inverting outputs and updating the hidden state (where the initial state $s_0$ is known and constant):

$$x_i = y^{-1} \left( r_\alpha^{-1} \left( z_i^{(r)} \right) - w^T s_{i-1} - b \right), \quad s_i = r \left( ux_i + v^T s_{i-1} + a \right). \tag{10}$$

Furthermore, the determinant of the Jacobian for (eq. 9) is the product of diagonal terms:

$$\det \frac{dz}{dx} = y^d \prod_{i=1}^{d} r'_\alpha \left( yx_i + w^T s_{i-1} + b \right), \tag{11}$$

where $r'_\alpha(t) = \mathbb{I}\{t > 0\} + \alpha \mathbb{I}\{t < 0\}$.

**Recurrent Shift Transformation:** It is worth noting that the rescaling brought on by the recurrent transformation effectively incurs a penalty through the log of the determinant (eq. 11). However, one can still perform a transformation that depends on the values of covariates through a shift operation. In particular, we propose an additive shift based on a recurrent function on prior dimensions:

$$z_i = x_i + m(s_{i-1}), \quad s_i = g(x_i, s_{i-1}), \tag{12}$$

where $g$ is some recurrent function for updating states, and $m$ is a fully connected network. Inversion proceeds as before:

$$x_i = z_i - m(s_{i-1}), \quad s_i = g(x_i, s_{i-1}), . \tag{13}$$

The Jacobian is again lower triangular, however due to the additive nature of (eq. 12), we have a unit diagonal. Thus, $\det \frac{dz}{dx} = 1$. One interpretation of this transformation is that one can shift the value of $x_k$ based on $x_{k-1}, x_{k-2}, \ldots$ for better conditional density estimation without any penalty coming from the determinant term in (eq. 7).

**Composing Transformations:** Lastly, we considering stacking (i.e. composing) several transformations $q = q^{(1)} \circ \ldots \circ q^{(T)}$ and renormalizing:

$$p(x_1, \ldots x_d) = \prod_{t=1}^{T} \left| \det \frac{dq^{(t)}}{dq^{(t-1)}} \right| \prod_{i=1}^{d} p \left( q_i(x) \, | \, q_{i-1}(x), \ldots, q_1(x) \right), \tag{14}$$

where we take $q^{(0)}$ to be $x$. We note that composing several transformations together allows one to leverage the respective strengths of each transformation. Moreover, inserting a reversal transformation $(x_1, \ldots, x_d \mapsto x_d, \ldots, x_1)$ in between transformations yields bidirectional relationships for several transformations.

## 2.3 COMBINED APPROACH

We combine the use of both transformations of variables and rich autoregressive models by: 1) writing the density of inputs, $p(x)$, as a normalized density of a transformation: $p(q(x))$ (eq. 14). Then we estimate the conditionals of $p(q(x))$ using an autoregressive model. I.e. to learn our model we minimize the negative log likelihood:

$$-\log p(x_1, \ldots x_d) = -\sum_{t=1}^{T} \log \left| \det \frac{\mathrm{d}q^{(t)}}{\mathrm{d}q^{(t-1)}} \right| - \sum_{i=1}^{d} \log p\left(q_i(x) \,|\, h_i\right), \qquad (15)$$

which is obtained by substituting (eq. 2) into (eq. 14) with $h_i$ as defined in (eq. 4).

## 3 RELATED WORK

Nonparametric density estimation has been a well studied problem in statistics and machine learning (Wasserman, 2007). Unfortunately, nonparametric approaches like kernel density estimation suffer greatly from the curse of dimensionality and do not perform well when data does not have a small number of dimensions ($d \lesssim 3$). To alleviate this, several semiparametric approaches have been explored. Such approaches include forest density estimation (Liu et al., 2011), which assumes that the data has a forest (i.e. a collection of trees) structured graph. This assumption leads to a density which factorizes in a first order Markovian fashion through a tree traversal of the graph. Another common semiparametric approach is to use a nonparanormal type model (Liu et al., 2009). This approach uses a Gaussian copula with a rank-based transformation and a sparse precision matrix. While both approaches are well-understood theoretically, their strong assumptions often lead to inflexible models.

In order to provide greater flexibility with semiparametric models, recent work has employed deep learning for density estimation. The use of neural networks for density estimation dates back to early work by Bishop (1994) and has seen success in areas like speech (Zen & Senior, 2014; Uria, 2015), music (Boulanger-Lewandowski et al., 2012), etc.. Typically such approaches use a network to learn to parameters of a parametric model for data. Recent work has also explored the application of deep learning to build density estimates in image data (Oord et al., 2016; Dinh et al., 2016). However, such approaches are heavily reliant on exploiting structure in neighboring pixels, often subsampling, reshaping or re-ordering data, and using convolutions to take advantage of neighboring correlations. Modern approaches for general density estimation in real-valued data include Uria et al. (2013; 2016); Germain et al. (2015); Gregor et al. (2014); Dinh et al. (2014); Kingma et al. (2016); Papamakarios et al. (2017).

NADE (Uria et al., 2013) is an RBM-inspired density estimator with a weight-sharing scheme across conditional densities on covariates. It may be written as a special case of LAM (eq. 5) with:

$$q_i\left(x_{i-1}, \ldots, x_1\right) = W_{<i} x_{<i} + b, \qquad (16)$$

where $W_{<i} \in \mathbb{R}^{p \times i-1}$ is the weight matrix compose of the first $i-1$ columns of a shared matrix $W = (w_1, \ldots w_d)$. We note also that LAM and NADE models are both related to fully visible sigmoid belief networks (Frey, 1998; Neal, 1992).

Even though the weight-sharing scheme in (eq. 16) reduces the number of parameters, it also greatly limits the types of distributions one can model. Roughly speaking, the NADE weight-sharing scheme makes it difficult to adjust conditional distributions when expanding the conditioning set with a covariate that has a small information gain. We illustrate these kinds of limitations with a simple example. Consider the following 3-dimensional distribution:

$$x_1 \sim \mathcal{N}(0, 1), \quad x_2 \sim \mathcal{N}(\mathrm{sign}(x_1), \epsilon), \quad x_3 \sim \mathcal{N}\left(\mathbb{I}\left\{|x_1| < C_{0.5}\right\}, \epsilon\right) \qquad (17)$$

where $C_{0.5}$ is the $50\%$ confidence interval of a standard Gaussian distribution, and $\epsilon > 0$ is some small constant. That is, $x_2$, and $x_3$ are marginally distributed as an equi-weighted bimodel mixture of Gaussian with means $-1, 1$ and $0, 1$, respectively. Due to NADE's weight-sharing linear model, it will be difficult to adjust $h_2$ and $h_3$ jointly to correctly model $x_2$ and $x_3$ respectively. However, given their additional flexibility, both LAM and RAM are able to adjust hidden states to remember and transform features as needed.

NICE models assume that data is drawn from a latent independent Gaussian space and transformed (Dinh et al., 2014). The transformation uses several "additive coupling" shifting transformations on

the second half of dimensions, using the first half of dimensions. That is, additive coupling proceeds by splitting inputs into halves $x = (x_{<d/2}, x_{\geq d/2})$, and transforming the second half as an additive function of the first half:

$$z = \big(x_{<d/2},\, x_{\geq d/2} + m(x_{<d/2})\big), \tag{18}$$

where $m(\cdot)$ is the output of a fully connected network. Inversion is simply a matter of subtraction $x = \big(z_{<d/2},\, z_{\geq d/2} - m(z_{<d/2})\big)$. The full transformation is the result of stacking several of these additive coupling layers together followed by a final rescaling operation. Furthermore, as with the RNN shift transformation, the additive nature of (eq. 18) yields a simple determinant, $\det \frac{\mathrm{d}z}{\mathrm{d}x} = 1$.

We also note that are several methods for obtaining samples from an unknown distribution that by-pass density estimation. For instance, generative adversarial networks (GANs) apply a (typically noninvertible) transformation of variables to a base distribution by optimizing a minimax loss over a discrimator and the transformation (Goodfellow, 2016). Furthermore, one can also obtain samples with only limited information about the density of interest. For example, if one has an unnormalized pdf, one may use Markov chain Monte Carlo (MCMC) methods to obtain samples (Neal, 1993).

## 4    EXPERIMENTS

We compare models using several experiments on synthetic and real-world datasets. First, we compute the average log likelihoods on test data. Then, to gain further context of the efficacy of models, we also use their density estimates for anomaly detection, where we take low density instances to be outliers. Moreover, we look at an illustrative MNIST image modeling task.

We study the performance of various combinations of conditional models and transformation. That is, we consider various models for the conditionals $p\big(q_i(x)\,\big|\,h_i\big)$ and various transformations $q(\cdot)$ (eq. 15). In particular the following conditional models were considered: `LAM`, `RAM`, `Tied`, `MultiInd`, and `SingleInd`. Here, `LAM`, `RAM`, and `Tied` are as described in equations (eq. 5), (eq. 6), and (eq. 16), respectively. `MultiInd` takes $p\big(q_i(x)\,\big|\,h_i\big)$ to be $p\big(q_i(x)\,\big|\,\mathrm{MM}(\theta_i)\big)$, that is we shall use $d$ distinct independent mixtures to model the transformed covariates. Similarly, `SingleInd` takes $p\big(q_i(x)\,\big|\,h_i\big)$ to be $p\big(q_i(x)\big)$, the density of a standard single component. Moreover, we considered the following transformations: `None`, `RNN`, `2xRNN`, `4xAdd+Re`, `4xSRNN+Re`, `RNN+4xAdd+Re`, and `RNN+4xSRNN+Re`. `None` indicates that no transformation of variables was performed. `RNN` and `2xRNN` performs a single recurrent transformation (eq. 9), and two recurrent transformations with a reversal permutation in between, respectively. Following (Dinh et al., 2014), `4xAdd+Re` performs four additive coupling transformations (eq. 18) with reversal permutations in between followed by a final element-wise rescaling: $x \mapsto x * \exp(s)$, where $s$ is a learned variable. Similarly, `4xSRNN+Re`, performs four recurrent shift transformations (eq. 12) with reversal permutations in between, followed by an element-wise rescaling. `RNN+4xAdd+Re`, and `RNN+4xSRNN+Re` are as before, but performing an initial recurrent transformation. Furthermore, we also considered performing an initial linear transformation (eq. 8). We flag this by prepending an `L` to the transformation; e.g. `L RNN` denotes a linear transformation followed by a recurrent transformation.

Models were implemented in Tensorflow (Abadi et al., 2016). Both RAM conditional models as well as the RNN shift transformation make use of the standard `GRUCell` GRU implementation[1]. We take the mixture models of conditionals (eq. 2) to be mixtures of 40 Gaussians. We optimize all models using the `AdamOptimizer` (Kingma & Ba, 2014) with an initial learning rate of $0.005$. Training consisted of $30\,000$ iterations, with mini-batches of size $256$. The learning rate was decreased by a factor of $0.1$, or $0.5$ (chosen via a validation set) every $5\,000$ iterations. Gradient clipping with a norm of 1 was used. After training, the best iteration according to the validation set loss was used to produce the test set mean log likelihoods.

### 4.1    SYNTHETIC

We perform a thorough empirical analysis over synthetic data. By carefully constructing data we will be able to pinpoint strengths and short-comings of conditional models and transformations. We study a dataset with a first-order Markovian structure, and one with a star-shaped structure; they are described below.

---

[1]Code will be made public upon publication.

### 4.1.1 MARKOVIAN DATA

First, we describe experiments performed on a synthetic dataset with a Markovian structure that features several exploitable correlations among covariates. The dataset is sampled as follows: $y_1, y_2, y_3 \sim \mathcal{N}(0, 1)$ and $y_i \,|\, y_{i-1}, \ldots, y_1 \sim f(i, y_1, y_2, y_3) + \epsilon_i$ for $i > 3$ where $\epsilon_i \sim \mathcal{N}(\epsilon_{i-1}, \sigma)$, $f(i, y_1, y_2, x_3) = y_1 \sin(y_2 g_i + y_3)$, and $g_i$'s are equi-spaced points on the unit interval. That is, instances are sampled using random draws of amplitude, frequency, and shift covariates $y_1, y_2, y_3$, which determine the mean of the other covariates, $y_1 \sin(y_2 g_i + y_3)$, stemming from function evaluations on a grid, and random noise $\epsilon_i$ with a Gaussian random walk. The resulting instances are easy to visualize, and contain many correlations among covariates (for instance, $y_4$ is highly informative of $y_5$). To test robustness to correlations from distant (by index) covariates, we observe covariates that are shuffled using a fixed permutation $\pi$ chosen ahead of time: $x = (y_{\pi_1}, \ldots, y_{\pi_d})$. We take $d = 32$, and the number of training instances to be $100\,000$.

We detail the mean log-likelihoods on a test set for TANs using various combinations of conditional models and transformations in Appendix, Table 6. Note that the performance of previous one-prong approaches that considered a complex conditional model with simple or no transformation and vice-versa are illustrated by `None & Tied` (NADE), `4xAdd+Re & SingleInd` (NICE) models, as well as by the entire row corresponding to `None` transformation and the `MultiInd` and `SingleInd` columns. We see that both `LAM` and `RAM` conditionals are providing most of the top models. We observe good samples from the best performing model (picked on validation dataset) as shown in Figure 3. Here we also observe relatively good performance stemming from `MultiInd` conditionals with more complex transformations.

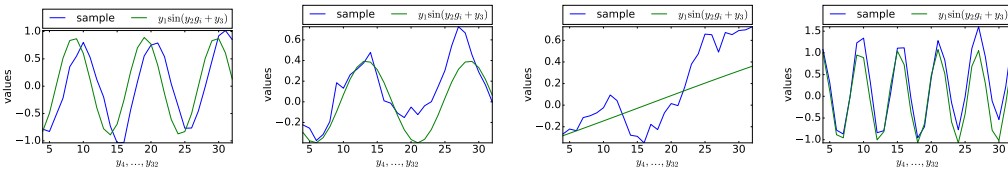

Figure 3: `RNN+4xSRNN+Re & RAM` model samples. Each plot shows a single sample. We plot the sample values of unpermuted dimensions $y_4, \ldots, y_{32} \,|\, y_1, y_2, y_3$ in blue and the expected value of these dimensions (i.e. without the Markovian noise) in green. One may see that the model is able to correctly capture both the sinusoidal and random walk behavior of our data.

### 4.1.2 STAR DATA

Next, we consider a dataset with a star-structured graphical model where fringe nodes are very uninformative with each-other. We divide the covariates into disjoint center and vertex sets $C = \{1, \ldots, 4\}$, $V = \{5, \ldots, d\}$ respectively. For center nodes $j \in C$, $y_j \sim \mathcal{N}(0, 1)$. Then, for $j \in V$, $y_j \sim \mathcal{N}(f_j(w_j^T y_C), \sigma)$ where $f_j$ is a fixed step function with 32 intervals, $w_j \in \mathbb{R}^4$ is a fixed vector, and $y_C = (y_1, y_2, y_3, y_4)$. We note that this dataset poses a difficult density estimation problem since the distribution of each of the fringe vertices will be considerably different from each other, the fringe vertices are largely uninformative from one another, and the distribution of the fringe vertices are difficult to estimate without conditioning on all the center nodes. As before we observe covariates that are shuffled using a fixed permutation $\pi$ chosen ahead of time: $x = (y_{\pi_1}, \ldots, y_{\pi_d})$, with $d = 32$.

We detail the mean log-likelihoods on a test set for TANs using various combinations of conditional models and transformations in the Appendix, Table 7. Once more we observe that both `LAM` and `RAM` conditionals are providing most of the top models. In this dataset, however, simpler conditional methods are unable to model the data well, suggesting that the complicated dependencies need a two-prong TAN approach. We observe a similar pattern when learning over data with $d = 128$ (see Appendix, Table 8).

Table 1: Average test log-likelihood. For each dataset and each conditional model, top-2 transformations are selected using log-likelihoods on a validation set and their mean test log-likelihood are reported. ∗ denotes the best model for each dataset picked by validation. Largest values per dataset are shown in **bold**.

| Dataset | LAM | | RAM | | Tied | | MultiInd | | NADE | NICE |
|---|---|---|---|---|---|---|---|---|---|---|
| forrest
d= 10
N=286,048 | 2.389
L RNN+
4xAdd+Re | 2.297
L RNN+
4xSRNN+Re | **2.672∗**
L RNN+
4xAdd+Re | 2.443
L RNN+
4xSRNN+Re | 0.909
L 4xAdd+
Re | 0.857
RNN+
4xAdd+Re | 0.754
L 4xSRNN+
Re | 0.600
L RNN+
4xSRNN+Re | −0.653 | −0.492 |
| pendigits
d= 16
N=6,870 | **6.923∗**
None | 5.854
4xSRNN+
Re | 3.896
2xRNN | 3.911
None | 1.437
None | −2.299
L RNN | −5.010
4xAdd+
Re | −4.742
L RNN+
4xAdd+Re | 1.437 | −6.498 |
| susy
d= 18
N=5,000,000 | 17.673
L 4xAdd+
Re | 17.474
L RNN+
4xAdd+Re | **18.941∗**
L RNN+
4xSRNN+Re | 18.389
L RNN | 15.397
L 4xSRNN+
Re | 13.765
L RNN+
4xAdd+Re | 12.161
L 4xSRNN+
Re | 12.105
L RNN+
4xSRNN+Re | −5.721 | 4.245 |
| higgs
d= 28
N=11,000,000 | −3.396
L RNN+
4xAdd+Re | −3.756
L RNN+
4xSRNN+Re | **−0.340∗**
L RNN | −2.116
RNN+
4xAdd+Re | −8.052
L RNN+
4xAdd+Re | −8.006
L 4xAdd+
Re | −8.223
L 4xSRNN+
Re | −9.378
L RNN+
4xSRNN+Re | −13.883 | −15.138 |
| hepmass
d= 28
N=10,500,000 | 3.906
L RNN+
4xAdd+Re | 3.759
L RNN+
4xSRNN+Re | 4.935∗
RNN | **5.047**
L RNN | −0.239
L RNN+
4xAdd+Re | −0.863
RNN+
4xSRNN+Re | −5.747
L 4xSRNN+
Re | −6.091
4xSRNN+
Re | −4.948 | −11.387 |
| satimage2
d= 36
N=5,803 | −1.716
None | −7.728
RNN | **−0.550∗**
L 2xRNN | −0.773
L RNN | −2.137
2xRNN | −2.549
4xSRNN+
Re | −1.570
L None | −1.699
L 2xRNN | −9.296 | −17.977 |
| music
d= 90
N=515,345 | **−51.572∗**
L RNN+
4xAdd+Re | −52.617
L RNN+
4xSRNN+Re | −55.665
L 4xSRNN+
Re | −56.190
L RNN+
4xAdd+Re | −58.885
L RNN+
4xAdd+Re | −59.093
L RNN+
4xAdd+Re | −69.484
L RNN+
4xAdd+Re | −69.887
L 4xAdd+
Re | −98.047 | −83.524 |
| wordvecs
d= 300
N=3,000,000 | **−247.440∗**
L 4xAdd+
Re | −248.393
L 4xSRNN+
Re | −272.371
L 4xAdd+
Re | −275.508
L 2xRNN | −273.372
L 4xSRNN+
Re | −273.976
L RNN+
4xAdd+Re | −308.148
L RNN+
4xSRNN+Re | −308.735
L 4xSRNN+
Re | −278.789 | −374.563 |

## 4.2 Test-Data Log Likelihoods

We used multiple datasets from the UCI machine learning repository[2] and Stony Brook outlier detection datasets collection (ODDS)[3] to evaluate log-likelihoods on test data. Broadly, the datasets can be divided into: **Particle acceleration**: higgs, hepmass, and susy datasets where generated for high-energy physics experiments using Monte Carlo simulations; **Music**: The music dataset contains timbre features from the million song dataset of mostly commercial western song tracks from the year 1922 to 2011; (Bertin-Mahieux et al., 2011). **Word2Vec**: wordvecs consists of 3 million words from a Google News corpus. Each word represented as a 300 dimensional vector trained using a word2vec model[4]. **ODDS datasets**: We used several ODDS datasets–forrest, pendigits, satimage2. These are multivariate datasets from varied set of sources meant to provide a broad picture of performance across anomaly detection tasks. To not penalize models for low likelihoods on outliers in ODDS, we removed anomalies from test sets when reporting log-likelihoods.

As noted in (Dinh et al., 2014), data degeneracies and other corner-cases may lead to arbitrarily low negative log-likelihoods. In order to avoid such complications, we remove discrete features, standardized all datasets, and add independent Gaussian noise with a standard deviation of 0.01 to training sets.

We report average test log-likelihoods in Table 1. For each dataset and conditional model, we report the test log-likelihood of the top-2 transformations (picked on a validation dataset). We note that the best performing model on each dataset had either LAM or RAM conditionals. The tables detailing test log-likelihoods for all combinations of conditional models and transformations for each dataset may be found in the Appendix (see Tables 10-16). We also observe that L RNN+4xAdd+Re & LAM and L RNN & RAM are consistently among the top-10 picked models.

### 4.2.1 Further Comparisons

In order to provide further context for the performance of our TAN models, we performed additional real-world data experiments and compared to several state-of-the-art auto-regressive density estimation methods. We carefully followed Papamakarios et al. (2017) and code (MAF Git Repository) to ensure that we operated over the same instances and covariates for each of the datasets considered in Papamakarios et al. (2017). In Table 2, we show the average test log-likelihoods of the best TAN model selected using a validation set and compare to values reported by Papamakarios et al. (2017) for MADE (Germain et al., 2015), Real NVP (Dinh et al., 2016), and MAF (Papamakarios et al., 2017) methods for each dataset. Also, we compared to MAF MoG, which uses a MAF transformation

---

[2] http://archive.ics.uci.edu/ml/

[3] http://odds.cs.stonybrook.edu

[4] https://code.google.com/archive/p/word2vec/

of variables (Papamakarios et al., 2017) with a MADE MoG conditional model (Germain et al., 2015).

As can be seen, our TAN models are considerably outperforming other state-of-the-art methods across all datasets used by Papamakarios et al. (2017).

Table 2: Average test log-likelihood comparison of TANs with baselines MADE, Real NVP, MAF as reported by Papamakarios et al. (2017). Parenthesized numbers indicate number of transformations used. Standard errors with $2\sigma$ are shown. Largest values per dataset are shown in **bold**.

| | POWER | GAS | HEPMASS | MINIBOONE | BSDS300 |
|---|---|---|---|---|---|
| MADE | -3.08 ± 0.03 | 3.56 ± 0.04 | -20.98 ± 0.02 | -15.59 ± 0.50 | 148.85 ± 0.28 |
| MADE MoG | 0.40 ± 0.01 | 8.47 ± 0.02 | -15.15 ± 0.02 | -12.27 ± 0.47 | 153.71 ± 0.28 |
| Real NVP (5) | -0.02 ± 0.01 | 4.78 ± 1.80 | -19.62 ± 0.02 | -13.55 ± 0.49 | 152.97 ± 0.28 |
| Real NVP (10) | 0.17 ± 0.01 | 8.33 ± 0.14 | -18.71 ± 0.02 | -13.84 ± 0.52 | 153.28 ± 1.78 |
| MAF (5) | 0.14 ± 0.01 | 9.07 ± 0.02 | -17.70 ± 0.02 | -11.75 ± 0.44 | 155.69 ± 0.28 |
| MAF (10) | 0.24 ± 0.01 | 10.08 ± 0.02 | -17.73 ± 0.02 | -12.24 ± 0.45 | 154.93 ± 0.28 |
| MAF MoG (5) | 0.30 ± 0.01 | 9.59 ± 0.02 | -17.39 ± 0.02 | -11.68 ± 0.44 | 156.36 ± 0.28 |
| TAN | **0.48 ± 0.01** | **11.19 ± 0.02** | **−15.12 ± 0.02** | **−11.01 ± 0.48** | **157.03 ± 0.07** |
| | L RNN+4xAdd+Re & RAM | L RNN+4xSRNN+Re & RAM | L RNN & RAM | 4xSRNN+Re & RAM | L RNN+4xSRNN+Re & RAM |

## 4.3 ANOMALY DETECTION

Next, we apply density estimates to anomaly detection. Typically anomalies or outliers are data-points that are unlikely given a dataset. In terms of density estimations, such a task is framed by identifying which instances in a dataset have a low corresponding density. That is, we shall label an instance $x$, as an anomaly if $\hat{p}(x) \leq t$, where $t \geq 0$ is some threshold and $\hat{p}$ is the density estimate based on training data. Note that this approach is trained in an unsupervised fashion. However, each methods' density estimates were evaluated on test data with anomaly/non-anomaly labels on instances. We used thresholded log-likelihoods on the test set to compute precision and recall. We use the average-precision metric: avg-prec $= \sum_{k=1}^{N_{\text{test}}} \text{precision}_r (\text{recall}_r - \text{recall}_{r-1})$ where $\text{precision}_r = \frac{tp_r}{tp_r+fp_r}$, $\text{recall}_r = \frac{tp_r}{tp_r+fn_r}$, and $tp_r, fp_r, fn_r$ are true positive anomalies, false positives and false negative respectively among the bottom $r$ log-likelihood instances in test data. Our results are shown in Table 3. We see that RAM performs the best on all three datasets. Beyond providing another interesting use for our density estimates, seeing good performance in these outlier detection tasks further demonstrates that our models are learning semantically meaningful patterns.

Table 3: Average precision score on outlier detection datasets. For each dataset and conditional, the average precision corresponding to the top-2 best transformation model, picked using likelihood on a validation set, is shown. The best score for each dataset is in **bold**, the number of outliers is $O$.

| Dataset | LAM | | RAM | | Tied | | MultiInd | | NADE | NICE |
|---|---|---|---|---|---|---|---|---|---|---|
| forrest | 0.936 | 0.902 | **0.944** | **0.944** | 0.918 | 0.923 | 0.928 | 0.882 | 0.866 | 0.802 |
| O=2,747 | L RNN+ 4xAdd+Re | L RNN+ 4xSRNN+Re | L RNN+ 4xAdd+Re | L RNN+ 4xSRNN+Re | L 4xAdd+ Re | RNN+ 4xAdd+Re | L 4xSRNN+ Re | L RNN+ 4xSRNN+Re | | |
| pendigits | 0.930 | 0.956 | **0.981** | 0.918 | 0.919 | 0.916 | 0.927 | 0.915 | 0.919 | 0.933 |
| O=156 | None | 4xSRNN+ Re | 2xRNN | None | None | L RNN | 4xAdd+ Re | L RNN+ 4xAdd+Re | | |
| satimage2 | 0.986 | 0.987 | **0.990** | 0.947 | **0.990** | 0.989 | 0.975 | 0.969 | **0.990** | 0.981 |
| O=71 | None | RNN | L 2xRNN | L RNN | 2xRNN | 4xSRNN+ Re | L None | L 2xRNN | | |

## 4.4 IMAGE MODELING

For illustrative purposes, we consider modeling MNIST digits. In keeping with our focus of general data modeling, we treat each image as a flattened vector of 784 dimensions. Here we demonstrate that our proposed models can be used to model high dimensional data and produce coherent samples.

First, we model dequantized pixel values rescaled to the unit interval as described in (Dinh et al., 2014). Moreover, we also model the MNIST digits through a logit transformation of pixel values. That is,

we take the dequantized pixel values in the range $[0, 256]$, $y$ and model: $x = \text{logit}\left(\lambda + (1 - \lambda)\frac{y}{256}\right)$, with $\lambda = 0.05$. This transformation will lessen boundary effects and keep pixel values inside a valid range.

We ran experiments on MNIST using the two models (`L RNN+4xAdd+Re & LAM` and `L RNN & RAM`) that consistently appear in the top-10 in our previous experiments (see Tables 10-16 in Appendix). We observe test set average log-likelihoods and samples reported in bits per pixel in Table 4. Furthermore, we plot samples in Figure 4. We see that our models are able to capture the structure of MNIST digits, with very few artifacts in samples. This is also reflected in the likelihoods, which are comparable or better than state-of-the-art.

Table 4: Bits per pixel for models (lower is better). "(Unit)" denotes model with unit scale on pixels, and "(Logit)" denotes model with logit transformation on pixels. Standard errors with $2\sigma$ are shown.

| NICE (Dinh et al., 2014) (Unit) | Real NVP (Papamakarios et al., 2017) (Logit) | MADE (Papamakarios et al., 2017) (Logit) | LAM `L RNN+4xAdd+Re` (Unit) | RAM `L RNN` (Unit) | LAM `L RNN+4xAdd+Re` (Logit) | RAM `L RNN` (Logit) |
|---|---|---|---|---|---|---|
| $4.47 \pm 0.021$ | $1.93 \pm 0.01$ | $1.41 \pm 0.01$ | $2.27 \pm 0.013$ | $1.60 \pm 0.007$ | $2.12 \pm 0.01$ | $\mathbf{1.19 \pm 0.005}$ |

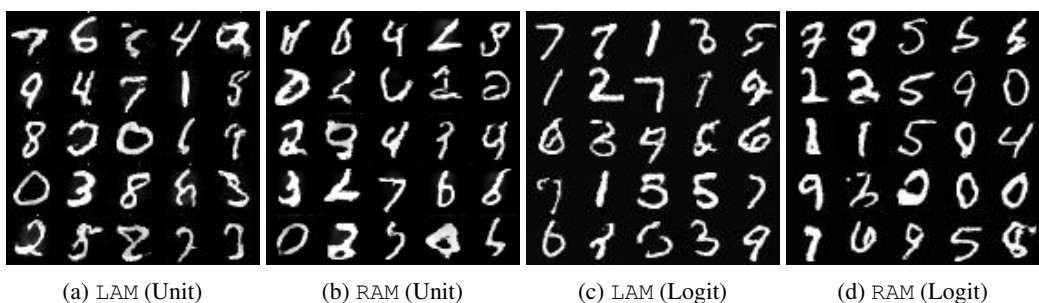

(a) `LAM` (Unit)      (b) `RAM` (Unit)      (c) `LAM` (Logit)      (d) `RAM` (Logit)

Figure 4: Samples from `L RNN+4xAdd+Re & LAM`, and `L RNN & RAM` models on unit scaled, and logit transformed pixels.

We also ran a TAN model on the CIFAR-10 dataset of $32 \times 32$ natural colored images. As before we focus on general data modeling, where we treat each image as a flattened vector. In Table 5, we compared our TAN model `RAM & L RNN` to several baselines reported in Papamakarios et al. (2017). Once again, we find that TANs are able to better model the CIFAR-10 dataset of images.

Table 5: Bits per pixel for models (lower is better) using logit transforms on CIFAR-10. MADE, Real NVP, and MAF values are as reported by Papamakarios et al. (2017). Standard errors with $2\sigma$ are shown.

| MADE | Real NVP | MAF | `RAM & L RNN` |
|---|---|---|---|
| $5.67 \pm 0.01$ | $4.53 \pm 0.01$ | $4.31 \pm 0.01$ | $\mathbf{3.98 \pm 0.01}$ |

## 5 DISCUSSION

We begin by noting the breadth of our proposed methods. As mentioned above, previous approaches considered a complex conditional model with a simple or no transformation and vice-versa. As such, some previous works have proposed a single new type of transformation, or a single new conditional model. Here, we propose multiple methods for transformations (linear, recurrent, and shift recurrent) and multiple autoregressive conditional models (LAM, and RAM). Furthermore, we consider the various combinations of transformations and autoregressive models, most of which constitute a novel TAN.

We draw several conclusions through our comprehensive empirical study. First, we consider our experiments on synthetic data. Methods that only consider complex transformations or condition models are illustrated in the entire row corresponding to the `None` transformation and the `MultiInd` and `SingleInd` columns, respectively. The performance of some of these models, which include

`None & Tied` (NADE), `4xAdd+Re & SingleInd` (NICE), was moderate on the Markovian data, however these one-prong approaches fail in the star dataset. Overall `LAM` and `RAM` methods provided considerable improvements, especially in the star dataset, where the flexibility of `LAM` made it possible to learn the widely different conditional probabilities present in the data.

Similarly, we observe that the best performing models in real-world datasets are those that incorporate a flexible transformation *and* conditional model. In fact, the best model (according to validation dataset) always has `LAM` or `RAM` autoregressive components. Hence, validation across models would always select one of these methods. In fact, $95\%$ of top-10 models (aggregated across all datasets) have a `LAM` and `RAM` conditional model (see Tables 10-16). It is interesting to see that many of these top models also contain a linear transformation. Of course, linear transformations of variables are common to most parametric models, however they have been under-explored in the context of autoregressive density estimation. Our methodology for efficiently learning linear transformations coupled with their strong empirical performance encourages their inclusion in autoregressive models.

Finally, we digest results over the real-world datasets by computing the percentage of the top likelihood achieved by each transformation $t$, and conditional model $m$, in dataset $D$: $s(t, m, D) = \exp(l_{t,m,D})/\max_{a,b} \exp(l_{a,b,D})$, where $l_{t,m,D}$ is the test log-likelihood for $t, m$ on $D$. We then average $S$ over the datasets: $S(t, m) = \frac{1}{T} \sum_D S(t, m, D)$, where $T$ is the total number of datasets. We show this score in the Appendix, Table 9. This table gives a summary of which models performed better (closer to the best performing model per dataset) over multiple datasets. We see that `RAM` conditional with `L RNN` transformation, and `LAM` conditional with `L RNN+4xAdd+Re` were the two best performers.

## 6 CONCLUSION

In conclusion, this work jointly leverages transformations of variables and autoregressive models, and proposes novel methods for both. We show a considerable improvement with our methods through a comprehensive study over both real world and synthetic data. Also, we illustrate the utility of our models in outlier detection and digit modeling tasks.

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

APPENDIX

Table 6: Held out test log-likelihoods for the Markovian dataset. The superscripts denote rankings of log-likelihoods on the validation dataset.

| Transformation | LAM | RAM | TIED | MultiInd | SingleInd |
|---|---|---|---|---|---|
| None | 14.319 | −29.950 | −0.612 | −41.472 | − − − |
| L None | **15.486$^{(9)}$** | 14.538 | 10.906 | 5.252 | −9.426 |
| RNN | 14.777 | −37.716 | 11.075 | −30.491 | −37.038 |
| L RNN | **15.658$^{(5)}$** | 10.354 | 10.910 | 5.370 | 3.310 |
| 2xRNN | 14.683 | 13.698 | 11.493 | −18.448 | −34.268 |
| L 2xRNN | **15.474$^{(8)}$** | **15.752$^{(3)}$** | 12.316 | 5.385 | 3.739 |
| 4xAdd+Re | 15.269 | 12.257 | 12.912 | 12.446 | 11.625 |
| L 4xAdd+Re | **15.683$^{(6)}$** | 12.594 | 13.845 | 12.768 | 12.069 |
| 4xSRNN+Re | 14.829 | 14.381 | 11.798 | 11.738 | 12.932 |
| L 4xSRNN+Re | 15.289 | **16.202$^{(1)}$** | 12.748 | **15.415$^{(10)}$** | 13.908 |
| RNN+4xAdd+Re | 15.171 | 12.991 | 14.455 | 11.467 | 10.382 |
| L RNN+4xAdd+Re | 15.078 | 12.655 | 14.415 | 12.886 | 12.315 |
| RNN+4xSRNN+Re | 14.968 | **16.216$^{(2)}$** | 12.590 | **15.589$^{(4)}$** | 14.231 |
| L RNN+4xSRNN+Re | 15.429 | **15.566$^{(7)}$** | 14.179 | 14.528 | 13.961 |

Table 7: Held out test log-likelihoods for star 32d dataset. The superscript denotes ranking of log-likelihood on cross validation dataset

| Transformation | LAM FC | RAM FC | TIED FC | MultiInd | SingleInd |
|---|---|---|---|---|---|
| None | −2.041 | 2.554 | −10.454 | −29.485 | − − − |
| L None | 5.454 | 8.247 | −7.858 | −26.988 | −38.952 |
| RNN | −1.276 | 2.762 | −6.292 | −25.946 | −41.275 |
| L RNN | 7.775 | 6.335 | −1.157 | −25.986 | −34.408 |
| 2xRNN | 3.705 | 8.032 | −0.565 | −25.100 | −38.490 |
| L 2xRNN | **14.878$^{(3)}$** | 9.946 | 0.901 | −23.772 | −33.075 |
| 4xAdd+Re | **13.278$^{(6)}$** | **11.561$^{(9)}$** | 7.146 | −16.740 | −21.332 |
| L 4xAdd+Re | **15.728$^{(2)}$** | **12.444$^{(7)}$** | 9.031 | −6.091 | −11.225 |
| 4xSRNN+Re | 3.496 | 8.429 | −1.380 | −15.590 | −23.712 |
| L 4xSRNN+Re | **16.042$^{(1)}$** | **9.939$^{(10)}$** | 5.598 | −12.530 | −16.889 |
| RNN+4xAdd+Re | **14.071$^{(5)}$** | **14.123$^{(4)}$** | 6.868 | −14.773 | −20.483 |
| L RNN+4xAdd+Re | **11.819$^{(8)}$** | 9.253 | 2.638 | −7.662 | −14.530 |
| RNN+4xSRNN+Re | −0.679 | 3.320 | −6.172 | −12.879 | −19.204 |
| L RNN+4xSRNN+Re | 7.433 | 7.324 | 3.554 | −10.427 | −15.243 |

Table 8: Held out test log-likelihood for Star 128d dataset.The superscript denotes ranking of log-likelihood on crossvalidation dataset

| Transformation | LAM FC | RAM FC | TIED FC | MultiInd | SingleInd |
|---|---|---|---|---|---|
| None | 15.671 | 15.895 | −83.115 | −128.238 | − − − |
| L None | 57.881 | −82.100 | −28.206 | −123.939 | −159.391 |
| RNN | 18.766 | 48.295 | −22.485 | −113.181 | −178.641 |
| L RNN | **66.070**$^{(9)}$ | −49.084 | 31.136 | −107.083 | −155.324 |
| 2xRNN | 27.295 | 45.834 | −11.930 | −113.210 | −178.331 |
| L 2xRNN | **85.681**$^{(3)}$ | −84.524 | 30.974 | −105.368 | −162.635 |
| 4xAdd+Re | **77.195**$^{(6)}$ | **61.947**$^{(10)}$ | 16.062 | −75.206 | −111.542 |
| L 4xAdd+Re | **88.837**$^{(1)}$ | −21.882 | 20.234 | −65.694 | −96.071 |
| 4xSRNN+Re | 33.577 | −98.796 | 3.256 | −88.912 | −98.936 |
| L 4xSRNN+Re | **86.375**$^{(2)}$ | **76.968**$^{(5)}$ | 33.481 | −85.590 | −93.086 |
| RNN+4xAdd+Re | **66.540**$^{(8)}$ | −57.861 | −16.277 | −75.491 | −114.729 |
| L RNN+4xAdd+Re | **80.063**$^{(4)}$ | 32.104 | 21.944 | −71.933 | −100.384 |
| RNN+4xSRNN+Re | 21.719 | −87.335 | −6.517 | −76.459 | −85.422 |
| L RNN+4xSRNN+Re | **72.463**$^{(7)}$ | 56.201 | 26.269 | −71.843 | −91.695 |

Table 9: Average performance percentage score for each model across all datasets. Note that this measure is not over a logarithmic space.

| Transformation | LAM | RAM | TIED | MultiInd | SingleInd | MAX |
|---|---|---|---|---|---|---|
| None | 0.218 | 0.118 | 0.006 | 0.000 | 0.000 | 0.218 |
| L None | 0.154 | 0.179 | 0.026 | 0.051 | 0.001 | 0.179 |
| RNN | 0.086 | 0.158 | 0.014 | 0.001 | 0.000 | 0.158 |
| L RNN | 0.173 | **0.540** | 0.014 | 0.040 | 0.013 | 0.540 |
| 2xRNN | 0.151 | 0.101 | 0.045 | 0.001 | 0.000 | 0.151 |
| L 2xRNN | 0.118 | 0.330 | 0.015 | 0.045 | 0.025 | 0.330 |
| 4xAdd+Re | 0.036 | 0.047 | 0.015 | 0.010 | 0.006 | 0.047 |
| L 4xAdd+Re | 0.153 | 0.096 | 0.025 | 0.014 | 0.009 | 0.153 |
| 4xSRNN+Re | 0.086 | 0.051 | 0.031 | 0.010 | 0.008 | 0.086 |
| L 4xSRNN+Re | 0.109 | 0.143 | 0.023 | 0.021 | 0.018 | 0.143 |
| RNN+4xAdd+Re | 0.121 | 0.096 | 0.023 | 0.011 | 0.011 | 0.121 |
| L RNN+4xAdd+Re | 0.336 | 0.165 | 0.024 | 0.016 | 0.013 | 0.336 |
| RNN+4xSRNN+Re | 0.102 | 0.151 | 0.017 | 0.012 | 0.014 | 0.151 |
| L RNN+4xSRNN+Re | 0.211 | 0.288 | 0.024 | 0.018 | 0.016 | 0.288 |
| MAX | 0.336 | 0.540 | 0.045 | 0.051 | 0.025 | |

Table 10: Held out test log-likelihood for `forest` dataset.The superscript denotes ranking of log-likelihood on crossvalidation dataset

| Transformation | LAM FC | RAM FC | TIED FC | MultiInd | SingleInd |
|---|---|---|---|---|---|
| None | 0.751 | −1.383 | −0.653 | −12.824 | − − − |
| L None | 1.910 | 1.834 | −0.243 | −7.665 | −11.062 |
| RNN | 1.395 | 0.053 | 0.221 | −5.130 | −15.983 |
| L RNN | **2.189**$^{(8)}$ | 1.747 | −0.087 | −4.001 | −5.807 |
| 2xRNN | 1.832 | 1.830 | 0.448 | −6.162 | −9.095 |
| L 2xRNN | **2.240**$^{(6)}$ | **2.432**$^{(3)}$ | 0.264 | −3.956 | −5.125 |
| 4xAdd+Re | 1.106 | 1.430 | 0.420 | −0.021 | −0.492 |
| L 4xAdd+Re | 2.043 | 1.979 | 0.909 | 0.365 | −0.088 |
| 4xSRNN+Re | 1.178 | 1.428 | 0.187 | −0.029 | −0.212 |
| L 4xSRNN+Re | **2.089**$^{(9)}$ | **2.061**$^{(10)}$ | 0.611 | 0.754 | 0.593 |
| RNN+4xAdd+Re | 1.962 | **2.226**$^{(7)}$ | 0.857 | 0.081 | 0.086 |
| L RNN+4xAdd+Re | **2.389**$^{(4)}$ | **2.672**$^{(1)}$ | 0.852 | 0.450 | 0.251 |
| RNN+4xSRNN+Re | 1.599 | 1.545 | 0.510 | 0.182 | 0.369 |
| L RNN+4xSRNN+Re | **2.297**$^{(5)}$ | **2.443**$^{(2)}$ | 0.804 | 0.600 | 0.480 |

Table 11: Held out test log-likelihood for `pendigits` dataset. The superscript denotes ranking of log-likelihood on crossvalidation dataset

| Transformation | LAM FC | RAM FC | TIED FC | MultiInd | SingleInd |
|---|---|---|---|---|---|
| None | $\mathbf{6.923^{(1)}}$ | $\mathbf{3.911^{(8)}}$ | $1.437$ | $-14.138$ | $---$ |
| L None | $\mathbf{4.104^{(9)}}$ | $2.911$ | $-2.872$ | $-9.997$ | $-15.617$ |
| RNN | $\mathbf{5.464^{(3)}}$ | $3.273$ | $-1.676$ | $-10.144$ | $-19.719$ |
| L RNN | $\mathbf{4.072^{(6)}}$ | $1.398$ | $-2.299$ | $-10.840$ | $-13.103$ |
| 2xRNN | $\mathbf{6.376^{(5)}}$ | $\mathbf{3.896^{(7)}}$ | $-4.002$ | $-12.132$ | $-16.576$ |
| L 2xRNN | $2.987$ | $0.871$ | $-3.977$ | $-10.890$ | $-12.711$ |
| 4xAdd+Re | $-1.924$ | $-3.087$ | $-3.172$ | $-5.010$ | $-6.498$ |
| L 4xAdd+Re | $-1.796$ | $-1.438$ | $-2.288$ | $-4.951$ | $-7.834$ |
| 4xSRNN+Re | $\mathbf{5.854^{(2)}}$ | $2.146$ | $-2.827$ | $-5.970$ | $-7.084$ |
| L 4xSRNN+Re | $3.758$ | $-1.020$ | $-3.370$ | $-5.885$ | $-12.978$ |
| RNN+4xAdd+Re | $-2.357$ | $-2.869$ | $-2.187$ | $-5.454$ | $-8.053$ |
| L RNN+4xAdd+Re | $-2.687$ | $-2.103$ | $-2.185$ | $-4.742$ | $-6.941$ |
| RNN+4xSRNN+Re | $\mathbf{5.207^{(4)}}$ | $2.425$ | $-2.126$ | $-5.147$ | $-8.859$ |
| L RNN+4xSRNN+Re | $\mathbf{3.466^{(10)}}$ | $0.496$ | $-2.761$ | $-7.205$ | $-13.897$ |

Table 12: Held out test log-likelihood for `susy` dataset.The superscript denotes ranking of log-likelihood on crossvalidation dataset

| Transformation | LAM FC | RAM FC | TIED FC | MultiInd | SingleInd |
|---|---|---|---|---|---|
| None | $9.736$ | $-14.821$ | $-5.721$ | $-21.369$ | $---$ |
| L None | $15.731$ | $\mathbf{16.930^{(8)}}$ | $6.410$ | $-8.846$ | $-17.130$ |
| RNN | $12.784$ | $3.347$ | $6.114$ | $-18.575$ | $-44.273$ |
| L RNN | $16.381$ | $\mathbf{18.389^{(2)}}$ | $6.772$ | $-5.744$ | $-11.489$ |
| 2xRNN | $11.052$ | $14.362$ | $3.595$ | $-16.478$ | $-33.126$ |
| L 2xRNN | $14.523$ | $\mathbf{17.373^{(7)}}$ | $10.687$ | $-6.884$ | $-10.420$ |
| 4xAdd+Re | $9.835$ | $8.033$ | $7.238$ | $6.031$ | $4.245$ |
| L 4xAdd+Re | $\mathbf{17.673^{(3)}}$ | $\mathbf{16.500^{(10)}}$ | $11.613$ | $10.941$ | $9.034$ |
| 4xSRNN+Re | $8.798$ | $13.235$ | $1.234$ | $6.936$ | $3.378$ |
| L 4xSRNN+Re | $14.242$ | $\mathbf{17.870^{(5)}}$ | $15.397$ | $12.161$ | $13.413$ |
| RNN+4xAdd+Re | $15.408$ | $12.480$ | $9.409$ | $7.619$ | $5.446$ |
| L RNN+4xAdd+Re | $\mathbf{17.474^{(6)}}$ | $16.376$ | $13.765$ | $10.951$ | $8.269$ |
| RNN+4xSRNN+Re | $14.066$ | $\mathbf{17.691^{(4)}}$ | $9.136$ | $10.088$ | $7.656$ |
| L RNN+4xSRNN+Re | $\mathbf{16.627^{(9)}}$ | $\mathbf{18.941^{(1)}}$ | $13.469$ | $12.105$ | $12.349$ |

Table 13: Held out test log-likelihood for `higgs` dataset.The superscript denotes ranking of log-likelihood on crossvalidation dataset

| Transformation | LAM FC | RAM FC | TIED FC | MultiInd | SingleInd |
|---|---|---|---|---|---|
| None | $-6.220$ | $-5.848$ | $-13.883$ | $-25.793$ | $---$ |
| L None | $\mathbf{-3.798^{(8)}}$ | $-10.651$ | $-9.084$ | $-16.025$ | $-36.051$ |
| RNN | $-5.800$ | $\mathbf{-2.600^{(3)}}$ | $-10.797$ | $-25.760$ | $-66.223$ |
| L RNN | $\mathbf{-3.975^{(9)}}$ | $\mathbf{-0.340^{(1)}}$ | $-8.574$ | $-18.607$ | $-32.753$ |
| 2xRNN | $-6.456$ | $-4.833$ | $-9.192$ | $-25.398$ | $-60.040$ |
| L 2xRNN | $-5.866$ | $\mathbf{-3.222^{(5)}}$ | $-8.216$ | $-16.083$ | $-30.730$ |
| 4xAdd+Re | $-6.502$ | $-10.491$ | $-9.356$ | $-13.678$ | $-15.138$ |
| L 4xAdd+Re | $-5.377$ | $-5.611$ | $-8.006$ | $-12.106$ | $-14.129$ |
| 4xSRNN+Re | $-7.422$ | $-6.863$ | $-11.033$ | $-11.878$ | $-12.182$ |
| L 4xSRNN+Re | $-5.999$ | $-9.329$ | $-8.474$ | $-8.223$ | $-8.926$ |
| RNN+4xAdd+Re | $\mathbf{-4.242^{(10)}}$ | $-4.804$ | $-9.187$ | $-12.321$ | $-15.261$ |
| L RNN+4xAdd+Re | $\mathbf{-3.396^{(6)}}$ | $\mathbf{-3.049^{(4)}}$ | $-8.052$ | $-12.246$ | $-13.765$ |
| RNN+4xSRNN+Re | $-5.262$ | $\mathbf{-2.116^{(2)}}$ | $-10.105$ | $-12.307$ | $-9.388$ |
| L RNN+4xSRNN+Re | $\mathbf{-3.756^{(7)}}$ | $-4.773$ | $-8.097$ | $-9.378$ | $-7.721$ |

Table 14: Held out test log-likelihood for `hepmass` dataset.The superscript denotes ranking of log-likelihood on crossvalidation dataset

| Transformation | LAM FC | RAM FC | TIED FC | MultiInd | SingleInd |
|---|---|---|---|---|---|
| None | $2.328$ | $\mathbf{3.710^{(6)}}$ | $-4.948$ | $-19.771$ | $---$ |
| L None | $\mathbf{3.570^{(7)}}$ | $2.517$ | $-4.052$ | $-9.266$ | $-35.042$ |
| RNN | $2.088$ | $\mathbf{4.935^{(1)}}$ | $-1.639$ | $-19.851$ | $-47.686$ |
| L RNN | $\mathbf{2.869^{(10)}}$ | $\mathbf{5.047^{(2)}}$ | $-2.920$ | $-16.032$ | $-30.210$ |
| 2xRNN | $1.774$ | $0.902$ | $-1.909$ | $-15.440$ | $-36.754$ |
| L 2xRNN | $2.053$ | $\mathbf{3.680^{(5)}}$ | $-2.150$ | $-15.457$ | $-24.079$ |
| 4xAdd+Re | $1.678$ | $1.873$ | $-4.046$ | $-9.117$ | $-11.387$ |
| L 4xAdd+Re | $1.961$ | $2.543$ | $-2.259$ | $-6.907$ | $-9.275$ |
| 4xSRNN+Re | $1.443$ | $2.156$ | $-2.904$ | $-6.091$ | $-7.186$ |
| L 4xSRNN+Re | $2.072$ | $2.730$ | $-3.014$ | $-5.747$ | $-6.245$ |
| RNN+4xAdd+Re | $2.817$ | $0.912$ | $-2.514$ | $-6.003$ | $-9.284$ |
| L RNN+4xAdd+Re | $\mathbf{3.906^{(3)}}$ | $-1.869$ | $-3.847$ | $-6.339$ | $-9.103$ |
| RNN+4xSRNN+Re | $2.663$ | $\mathbf{3.586^{(8)}}$ | $-0.863$ | $-7.146$ | $-3.939$ |
| L RNN+4xSRNN+Re | $\mathbf{3.759^{(4)}}$ | $\mathbf{3.487^{(9)}}$ | $-0.239$ | $-7.522$ | $-6.102$ |

Table 15: Held out test log-likelihood for `satimage2` dataset.The superscript denotes ranking of log-likelihood on crossvalidation dataset

| Transformation | LAM FC | RAM FC | TIED FC | MultiInd | SingleInd |
|---|---|---|---|---|---|
| None | $\mathbf{-1.716^{(9)}}$ | $\mathbf{-1.257^{(3)}}$ | $-9.296$ | $-50.507$ | $---$ |
| L None | $-20.164$ | $\mathbf{-1.079^{(4)}}$ | $-2.635$ | $\mathbf{-1.570^{(5)}}$ | $-5.972$ |
| RNN | $-7.728$ | $-4.949$ | $-5.466$ | $-6.047$ | $-16.521$ |
| L RNN | $-31.296$ | $\mathbf{-0.773^{(2)}}$ | $-3.944$ | $\mathbf{-1.824^{(8)}}$ | $-2.977$ |
| 2xRNN | $-12.283$ | $\mathbf{-2.193^{(7)}}$ | $-2.137$ | $-5.447$ | $-8.075$ |
| L 2xRNN | $-20.968$ | $\mathbf{-0.550^{(1)}}$ | $-5.140$ | $\mathbf{-1.699^{(6)}}$ | $\mathbf{-2.276^{(10)}}$ |
| 4xAdd+Re | $-19.931$ | $-7.539$ | $-11.826$ | $-18.901$ | $-17.977$ |
| L 4xAdd+Re | $-21.128$ | $-9.944$ | $-12.336$ | $-21.677$ | $-24.070$ |
| 4xSRNN+Re | $-7.519$ | $-11.368$ | $-2.549$ | $-7.730$ | $-7.232$ |
| L 4xSRNN+Re | $-18.170$ | $-7.709$ | $-5.533$ | $-17.085$ | $-15.347$ |
| RNN+4xAdd+Re | $-19.278$ | $-11.789$ | $-12.837$ | $-21.249$ | $-22.786$ |
| L RNN+4xAdd+Re | $-20.899$ | $-12.949$ | $-12.867$ | $-26.164$ | $-28.302$ |
| RNN+4xSRNN+Re | $-13.476$ | $-3.951$ | $-6.284$ | $-15.025$ | $-16.443$ |
| L RNN+4xSRNN+Re | $-20.179$ | $-12.128$ | $-7.258$ | $-18.065$ | $-18.125$ |

Table 16: Held out test log-likelihood for `music` dataset.The superscript denotes ranking of log-likelihood on crossvalidation dataset

| Transformation | LAM FC | RAM FC | TIED FC | MultiInd | SingleInd |
|---|---|---|---|---|---|
| None | $-57.873$ | $-97.925$ | $-98.047$ | $-113.099$ | $---$ |
| L None | $\mathbf{-52.954^{(4)}}$ | $-74.220$ | $-72.441$ | $-82.866$ | $-104.287$ |
| RNN | $\mathbf{-54.933^{(10)}}$ | $-80.436$ | $-74.361$ | $-106.219$ | $-144.735$ |
| L RNN | $\mathbf{-52.710^{(3)}}$ | $-59.815$ | $-66.536$ | $-82.731$ | $-98.813$ |
| 2xRNN | $-56.958$ | $-85.359$ | $-77.456$ | $-104.440$ | $-133.898$ |
| L 2xRNN | $\mathbf{-53.956^{(8)}}$ | $-57.611$ | $-65.016$ | $-82.678$ | $-96.542$ |
| 4xAdd+Re | $-56.349$ | $-69.302$ | $-67.064$ | $-73.886$ | $-83.524$ |
| L 4xAdd+Re | $\mathbf{-53.169^{(5)}}$ | $-59.282$ | $-59.093$ | $-69.887$ | $-79.330$ |
| 4xSRNN+Re | $-57.670$ | $-68.116$ | $-74.006$ | $-78.032$ | $-121.197$ |
| L 4xSRNN+Re | $\mathbf{-53.879^{(7)}}$ | $-55.665$ | $-63.894$ | $-77.564$ | $-81.188$ |
| RNN+4xAdd+Re | $\mathbf{-53.177^{(6)}}$ | $-67.377$ | $-63.372$ | $-73.882$ | $-84.032$ |
| L RNN+4xAdd+Re | $\mathbf{-51.572^{(1)}}$ | $-56.190$ | $-58.885$ | $-69.484$ | $-79.555$ |
| RNN+4xSRNN+Re | $\mathbf{-54.065^{(9)}}$ | $-61.204$ | $-76.437$ | $-71.814$ | $-81.087$ |
| L RNN+4xSRNN+Re | $\mathbf{-52.617^{(2)}}$ | $-68.756$ | $-65.061$ | $-83.292$ | $-78.997$ |

Table 17: Held out test log-likelihood for `wordvecs` dataset.The superscript denotes ranking of log-likelihood on validation dataset. Due to time constraints only models with linear transformations were trained.

| Transformation | LAM FC | RAM FC | TIED FC | MultiInd | SingleInd |
|---|---|---|---|---|---|
| L None | $\mathbf{-252.659^{(6)}}$ | $-279.788$ | $-278.789$ | $-332.474$ | $-387.341$ |
| L RNN | $\mathbf{-252.894^{(7)}}$ | $-278.795$ | $-278.663$ | $-332.689$ | $-386.700$ |
| L 2xRNN | $\mathbf{-250.285^{(4)}}$ | $-275.508$ | $-277.848$ | $-333.234$ | $-386.649$ |
| L 4xAdd+Re | $\mathbf{-247.440^{(1)}}$ | $\mathbf{-272.371^{(8)}}$ | $-274.205$ | $-331.148$ | $-374.563$ |
| L 4xSRNN+Re | $\mathbf{-248.393^{(2)}}$ | $-300.666$ | $\mathbf{-273.372^{(9)}}$ | $-308.735$ | $0.000$ |
| L RNN+4xAdd+Re | $\mathbf{-249.980^{(3)}}$ | $-280.938$ | $\mathbf{-273.976^{(10)}}$ | $-331.316$ | $-380.031$ |
| L RNN+4xSRNN+Re | $\mathbf{-251.468^{(5)}}$ | $-280.325$ | $-274.082$ | $-308.148$ | $-395.084$ |