# OpenReview forum: "Transformation Autoregressive Networks"
_ICLR.cc/2018/Conference — Reject_

### Official Review · AnonReviewer2 · 2017-11-27
**Comparison with MAF missing in several Tables**

**Rating:** 5
**Confidence:** 3

**Review:**

The authors propose to combine nonlinear bijective transformations and flexible density models for density estimation. In terms of bijective change of variables transformations, they propose linear triangular transformations and recurrent transformations. They also propose to use as base transformation an autoregressive distribution with mixture of gaussians emissions.
Comparing with the Masked Autoregressive Flows (Papamakarios et al., 2017) paper, it seems that the true difference is using the linear autoregressive transformation (LAM) and recurrent autoregressive transformation (RAM), already present in the Inverse Autoregressive Flow (Kingma et al., 2016) paper they cite, instead of the masked feedforward architecture used Papamakarios et al. (2017).
Given that, the most important part of the paper would be to demonstrate how it performs compared to Masked Autoregressive Flows. A comparison with MAF/MADE is lacking in Table 1 and 2. Nonetheless, the comparison between models in flexible density models, change of variables transformations and combinations of both remain relevant.

Diederik P. Kingma, Tim Salimans, Rafal Józefowicz, Xi Chen, Ilya Sutskever, Max Welling: Improving Variational Autoencoders with Inverse Autoregressive Flow. NIPS 2016
George Papamakarios, Theo Pavlakou, Iain Murray: Masked Autoregressive Flow for Density Estimation. NIPS 2017

---

> ### Author Response · Authors · 2017-12-08
> **Thank you, please see general comment above**
>
> Thank you for your comments and suggestions. Please see our general reply where we address comparisons to MAF. Also, we would like to emphasize that LAM and RAM components are not deterministic transformations as IAF and MAF, but are modeling the conditional distribution of covariates using a mixture of gaussians.

---

> > ### Comment · AnonReviewer2 · 2017-12-31
> > **Difference with MAF**
> >
> > The MAF paper uses MADE which already models conditional distributions using Mixture of Gaussians, hence the "MAF MoG" label in the experiments (which you copied).
> > Your contributions need to be better emphasized.

---

> > > ### Author Response · Authors · 2017-12-31
> > > **Clarification**
> > >
> > > This paper introduces *multiple* new methods for both a conditional model for factors of the chain rule and a transformation of variables: the LAM conditional model, the RAM conditional model, the LU linear transformation, the recurrent transformation, and recurrent shift transformation.
> > >
> > > Our extensive empirical study shows the fundamental result that modern density estimation methods should employ *both* a flexible conditional model and a flexible transformation (e.g. using a MAF transform with MADE MoG conditional). Moreover, these new comparisons of TANs to MADE, Real NVP, MAF, and MAF MoG methods show that the combination of our proposed transformations and conditional models are superior.
> > >
> > > We have better emphasized our contributions in our revised introduction section (see page one).

---

### Official Review · AnonReviewer3 · 2017-11-27
**Incremental work with unclear contribution**

**Rating:** 5
**Confidence:** 2

**Review:**

This paper offers an extension to density estimation networks that makes them better able to learn dependencies between covariates of a distribution.

This work does not seem particularly original as applying transformations to input is done in most AR estimators.

Unfortunately, it's not clear if the work is better than the state-of-the-art. Most results in the paper are comparisons of toy conditional models. The paper does not compare to work for example from Papamakarios et al. on the same datasets. The one Table that lists other work showed LAM and RAM to be comparable. Many of the experiments are on synthetic results, and the paper would have benefited from concentrating on more real-world datasets.

---

> ### Author Response · Authors · 2017-12-08
> **Thank you, please see general comment above**
>
> Thank you for your comments and suggestions. Please see our general reply where we address comparisons to MAF and our contributions. Also, we would like to emphasize that we have increased the number of real-world datasets used to evaluate performance from 9 to 14.

---

### Official Review · AnonReviewer1 · 2017-11-27
**Solid description of a general class of autoregressive density estimation models with potential utility**

**Rating:** 8
**Confidence:** 4

**Review:**

This paper is well constructed and written. It consists of a number of broad ideas regarding density estimation using transformations of autoregressive networks. Specifically, the authors examine models involving linear maps from past states (LAM) and recurrence relationships (RAM).

The critical insight is that the hidden states in the LAM are not coupled allowing considerable flexibility between consecutive conditional distributions. This is at the expense of an increased number of parameters and a lack of information sharing. In contrast, the RAM transfers information between conditional densities via the coupled hidden states allowing for more constrained smooth transitions.

The authors then explored a variety of transformations designed to increase the expressiveness of LAM and RAM. The authors importantly note that one important restriction on the class of transformations is the ability to evaluate the Jacobian of the transformation efficiently. A composite of transformations coupled with the LAM/RAM networks provides a highly expressive model for modelling arbitrary joint densities but retaining interpretable conditional structure.

There is a rich variety of synthetic and real data studies which demonstrate that LAM and RAM consistently rank amongst the top models demonstrating potential utility for this class of models.

Whilst the paper provides no definitive solutions, this is not the point of the work which seeks to provide a description of a general class of potentially useful models.

---

> ### Author Response · Authors · 2017-12-08
> **Thank you**
>
> Thank you for your time and insightful comments.

---

### Author Response · Authors · 2017-12-08
**Comprehensive comparisons with MAF, MADE, and Real NVP**

We would like to thank all the reviewers for their time and helpful comments.

We agree with reviewers that comparing to MAFs strengthens our paper. At the time of writing we were unaware that the work by Papamakarios et al. (2017) was to be published and hence did not extensively compare to those results; we now revise with added comparisons.

Working off of both the paper and code in https://github.com/gpapamak/maf, we carefully preprocessed the datasets found in (Papamakarios et al. 2017) to work over the same instances/covariates. As can be seen in Section 4.2.1 Table 2 and Section 4.4 Table 4, Table 5, we are considerably beating both MAF, MADE, and Real NVP models in every dataset used by Papamakarios et al. (2017) (POWER, GAS, HEPMASS, MINIBOONE, BSDS300, MNIST, CIFAR-10).

We also wish to reemphasize the extent of our contribution. Whereas many modern density estimation work will introduce a single new conditional model for factors of the chain rule or a single new transformation of variables, this paper introduces *multiple* new methods for each of these components: the LAM conditional model, the RAM conditional model, the LU linear transformation, the recurrent transformation, and recurrent shift transformation.

In addition, our extensive empirical study shows the fundamental result that modern density estimation methods should employ *both* a flexible conditional model and a flexible transformation. Our extensive original experiments coupled with these new comparisons of TANs to MADE, Real NVP, and MAF make a very strong case for using TANs for density estimations.

---

### Decision · Program_Chairs · 2018-01-29
**ICLR 2018 Conference Acceptance Decision**

**Decision:**

Reject

**Comment:**

This paper looks at  building new density estimation methods and new methods for tranformations and autoregressive models. The request from reviewers for comparison improves the paper. These models have seen a wide range of applications and have been highly successful, needing the added benefits shown and their potential impact to be expanded further.